# Comprehensive Analysis and Expression Profiling of PIN, AUX/LAX, and ABCB Auxin Transporter Gene Families in *Solanum tuberosum* under Phytohormone Stimuli and Abiotic Stresses

**DOI:** 10.3390/biology10020127

**Published:** 2021-02-05

**Authors:** Chenghui Yang, Dongdong Wang, Chao Zhang, Minghui Ye, Nana Kong, Haoli Ma, Qin Chen

**Affiliations:** 1State Key Laboratory of Crop Stress Biology for Arid Areas, College of Agronomy, Northwest A&F University, Yangling 712100, China; 2018060035@nwafu.edu.cn (C.Y.); dongdong-1025@hotmail.com (D.W.); zhangchao520@nwafu.edu.cn (C.Z.); yeminghui@nwafu.edu.cn (M.Y.); knnnwafu@163.com (N.K.); 2Department of Biological Repositories, Zhongnan Hospital of Wuhan University, Wuhan 430071, China; 3College of Food Science and Engineering, Northwest A&F University, Yangling 712100, China

**Keywords:** potato, auxin transporter, PIN, AUX/LAX, ABCB, abiotic stress

## Abstract

**Simple Summary:**

In this study, we provide comprehensive information on auxin transporter gene families in potato, including basic parameters, chromosomal distribution, phylogeny, co-expression network analysis, gene structure, tissue-specific expression patterns, subcellular localization, transcription analysis under exogenous hormone stimuli and abiotic stresses, and *cis*-regulatory element prediction. The responsiveness of auxin transporter family genes to auxin and polar auxin transport inhibitors implied their possible roles in auxin homoeostasis and redistribution. Additionally, the differential expression levels of auxin transporter family genes in response to abscisic acid and abiotic stresses suggested their specific adaptive mechanisms on tolerance to various environmental stimuli. Promoter *cis*-regulatory element description analyses indicated that a number of *cis*-regulatory elements within the promoters of auxin transporter genes in potato were targeted by relevant transcription factors to respond to diverse stresses. We are confident that our results provide a foundation for a better understanding of auxin transporters in potato, as we have demonstrated the biological significance of this family of genes in hormone signaling and adaption to environmental stresses.

**Abstract:**

Auxin is the only plant hormone that exhibits transport polarity mediated by three families: auxin resistant (AUX) 1/like AUX1 (LAX) influx carriers, pin-formed (PIN) efflux carriers, and ATP-binding cassette B (ABCB) influx/efflux carriers. Extensive studies about the biological functions of auxin transporter genes have been reported in model plants. Information regarding these genes in potato remains scarce. Here, we conducted a comprehensive analysis of auxin transporter gene families in potato to examine genomic distributions, phylogeny, co-expression analysis, gene structure and subcellular localization, and expression profiling using bioinformatics tools and qRT-PCR analysis. From these analyses, 5 StLAXs, 10 StPINs, and 22 StABCBs were identified in the potato genome and distributed in 10 of 18 gene modules correlating to the development of various tissues. Transient expression experiments indicated that three representative auxin transporters showed plasma membrane localizations. The responsiveness to auxin and auxin transport inhibitors implied their possible roles in mediating intercellular auxin homoeostasis and redistribution. The differential expression under abscisic acid and abiotic stresses indicated their specific adaptive mechanisms regulating tolerance to environmental stimuli. A large number of auxin-responsive and stress-related *cis*-elements within their promoters could account for their responsiveness to diverse stresses. Our study aimed to understand the biological significance of potato auxin transporters in hormone signaling and tolerance to environmental stresses.

## 1. Introduction

Auxin is a plant hormone that possesses multiple functions that regulate the plant growth and development primarily at the cellular level and in response to diverse environmental stimuli as well [1,2,3,4,5]. Auxin is primarily synthesized in leaf primordium, germinating seeds, root tips, and cambiums. The directional polar auxin transport (PAT) system distributes auxin to the targeted tissues to facilitate apical dominance, embryonic development, vascular tissue development, root meristem maintenance, and organ formation and positioning [6]. The cell–cell polar transport of auxin is mediated by proteins localized in the plasma membrane (PM), and these proteins are members of three distinct gene families: the auxin resistant (AUX) 1/like AUX1 (LAX) influx carriers [7], the pin-formed (PIN) efflux carriers [8], and the ATP-binding cassette B/Multidrug-resistance /P-glycoprotein (ABCB/MDR/PGP) efflux/condition transporters [9]. Unique polar transport of auxin forms auxin gradients that result in asymmetric distribution of PIN, AUX/LAX, and ABCB proteins across cells and tissues. Regulation of the uneven distribution of auxin within tissues and organs or throughout the entire plant body via auxin transporters provides an important strategy to execute the auxin functions in controlling various plant developmental processes and in response to stressful environments.

AUX/LAX influx carriers encompass four highly conversed family members, and AUX1 and LAX1–3 are characterized as multi membrane-spanning transmembrane proteins that facilitated the entry of auxin into cells [10]. The founder member AUX1 is initially identified from Arabidopsis and primarily functions to promote high-affinity cellular auxin uptake in the root tips [11]. The auxin insensitive1 (*aux1*) mutant exhibits an agravitropic phenotype that is observed in roots after treatment with the auxin influx carrier inhibitor 1-naphthoxyaceticacids (1-NOA), and the gravitropic response in this mutant can be restored by treatment with the membrane-permeable auxin 1-naphthaleneacetic acid (NAA) [12,13]. AtLAX2 participates in the vascular development of cotyledons, and disruption of this gene reinforces cell separation in the quiescent center (QC) and reduces the expression of the auxin response reporter DR5 [14]. LAX3 in combination with AUX1, appears to facilitate lateral root development by targeting the auxin-inducible expression of a selection of cell wall remodeling enzymes. The *lax3* mutation in Arabidopsis disrupts lateral root emergence, while the *aux1lax3* double mutant exhibits a reduced number of emerged lateral roots [15]. The AUX/LAX gene family has also been described in different monocotyledons and dicotyledons in response to hormonal and abiotic stress at the transcriptional level [16,17,18,19,20].

The PIN family is the most extensively studied of the auxin efflux carriers among the auxin transporters in plants, and these carriers play an essential role in PAT [21]. The PIN genes are first cloned in Arabidopsis, where eight members of this gene family are well characterized. PIN proteins are localized either on the PM (AtPIN1, -2, -3, -4 and -7) or in the endoplasmic reticulum (ER) (AtPIN5, AtPIN6, and AtPIN8), and corresponding classifications are further divided based on their different structures, which determine their distinct roles in polar intercellular transport and intracellular auxin homeostasis [22]. AtPIN1 is expressed in the vascular bundle of the root, in the inflorescence stem, and the developing organs [23,24]. Loss-of-function mutations in this gene result in defective floral organs and the formation of naked, pin-shaped inflorescences, fused leaves, and other shoot abnormalities [25]. Initially, three identified mutant alleles of PIN2 that possess a strong root agravitropic phenotype were named independently as ethylene-insensitive root1 (*eir1*), agravitropic1 (*agr1*), and wavy6 (*wav6*) [26,27,28]. AtPIN3 is symmetrically located in the root columella cells; however, this protein rapidly relocalizes laterally in response to gravity stimulation. Mutations in the Arabidopsis gene PIN3 result in reduced growth and tropic response, and apical hook formation is also altered in *pin3* mutants [29]. The functional involvement of AtPIN4 during organogenesis and embryogenesis has been also demonstrated. Disruption of AtPIN4 affects pattern formation in both the embryo and the seedling roots. Therefore, AtPIN4 plays an essential role in auxin maximization and redistribution within the root tip [30]. AtPIN7 is expressed in the basal lineage in the embryo and then later in the root tips, and this protein displays an expression pattern that is complementary to that of PIN1 [25]. These loss-of-function phenotypes demonstrate the crucial role of these proteins in these developmental processes.

AtPIN5 has been implicated in regulating intracellular auxin homoeostasis and metabolism. *pin5* loss- and gain-of-function mutants have been observed to be defective in root and hypocotyl growth [31]. ER localization of PIN6 during auxin homoeostasis was required for nectary auxin response, short stamen development, and auxin distribution during root organogenesis in Arabidopsis [32,33]. Aberrant expression or loss-of-function of PIN6 could interfere with multiple auxin-regulated growth functions, including shoot apical dominance, lateral root primordia development, adventitious root formation, root hair growth, and root waving, indicating that PIN6 acted as a crucial component of auxin homoeostasis [34]. AtPIN8 was preferentially expressed in male gametophytes, with a specific accumulation in pollen from microspore to mature pollen and during pollen germination [35]. The effects of loss or gain of function of PIN6, PIN8, and PIN5 were not limited to vein patterning and could extend to the modulation of intracellular auxin response levels [36].

The ATP-binding cassette (ABC) superfamily is a large and diverse group (A–H) of proteins, and over 100 ABC proteins have been identified to date in plants [37]. In the subfamily B that includes homologs of the mammalian MDRs/PGPs, six members of the ABCB transporter family in Arabidopsis (AtABCB1, AtABCB4, AtABCB14, AtABCB15, AtABCB19, and AtABCB21) have been reported to mediate cellular auxin transport or auxin derivatives. Of these transporters, AtABCB1, AtABCB4, and AtABCB19 were the best-characterized [38]. Since the first plant MDR-like gene (AtABCB1/PGP1/MDR1) was cloned from Arabidopsis, it has been reported that AtPGP1 was localized to the PM and that the corresponding gene was expressed in both the root and shoot apex [39,40]. Additionally, the loss of AtMDR1 resulted in epinastic cotyledons and reduced apical dominance. Thus, researchers have speculated that AtPGP1 might transport a growth-regulating molecule known as indole-3-acetic acid (IAA) from the shoot apex to influence the distribution of the hormone auxin during plant development [41]. AtABCB19 has been identified as a stable PM protein that mainly localized to the vascular tissues of the hypocotyl and to the stelae of the root. Defects in PGP19 impaired basipetal auxin transport and resulted in abnormalities in the straight growth of hypocotyl [42]. Moreover, ABCB19, coordinated with PIN1, functioned to direct auxin flow from the shoot apex to a maximum in roots [43]. AtABCB4/PGP4 was a root-specific transporter that was predominantly expressed during early root development, and the expression of this transporter has been observed within the root elongation zone and lateral root, and during root hair initiation. *Atpgp4* mutants exhibited several root phenotypes, such as abnormal lateral root initiation, enhanced root hair elongation, and reduced basipetal auxin transport in roots. These phenotypes suggested the direct involvement of AtPGP4 in auxin homoeostasis within the root [44].

Auxin transporter genes have been widely studied throughout the plant kingdom, including monocotyledons such as *Oryza sativa*, *Sorghum bicolor*, and *Zea mays*, and *Arabidopsis thaliana*, *Populus trichocarpa*, *Citrullus lanatus*, *Medicago truncatula*, and *Brassica rapa* L., etc., belonging to dicotyledons as well [45,46,47]. However, a systematic study of auxin transporter genes in potato is lacking. Given the important roles of auxin transporter proteins during plant growth and development and in response to diverse environmental stimuli, we provide here the first comprehensive information detailing the StLAX, StPIN, and StABCB auxin transporter gene families in potato, and we systematically analyze their genomic distributions, gene structures, phylogenic relationships, co-expression analysis, three-dimensional (3D) structure prediction, and subcellular localization and expression profiles. In this study, we emphasized the distinctive spatio–temporal expression patterns of putative StLAX, StPIN, and StABCB genes in response to phytohormone stimuli and abiotic stress. Two auxin transport inhibitors (PATIs) were used to screen for candidate members of auxin transporter gene families responsible for auxin transport. Additionally, *cis*-regulatory element analysis was incorporated to further examine their expression profiling. Our study aimed to provide a foundation for the further exploration of the biological functions of auxin transporter genes.

## 2. Materials and Methods

### 2.1. Identification of AUX/LAX, PIN, and ABCB Auxin Transporter Family Genes in Potato

To identify the putative AUX/LAX and ABCB genes in *Solanum tuberosum*, the available protein sequence data (DM_v3.4_pep_nonredundant) of potato were downloaded from the Potato Genome Sequencing Consortium (PGSC) [48]. The gene identifiers of AtAUX/LAX and AtABCB were obtained from Balzan et al. [45], and sequences of AtAUX/LAX and AtABCB genes retrieved from phytozome 12.1.6 (https://phytozome.jgi.doe.gov/pz/portal.html accessed on 31 May 2020) served as queries to perform the BLAST searches. Then, the Hidden Markov Model (HMM) was used to identify target sequences obtained from the *S. tuberosum* genome. Pfam 01490 (transmembrane amino acid transporter protein) was used for the AUX/LAX family identification and Pfam 00005 (ABC transporter) and Pfam 00664 (ABC transporter transmembrane region) were used for the ABCB family. The remaining sequences were checked for further membrane transport protein domains using InterProScan Sequence Search (http://www.ebi.ac.uk/Tools/pfa/iprscan/ accessed on 31 May 2020). All identified AUX/LAX and ABCB proteins among *S. tuberosum* genome together with 10 StPIN proteins based on Efstathios R’s publication [49] were preserved for downstream analysis. Moreover, their information of molecular weights (MW) and isoelectric points (pI) were calculated by Pepstats (https://www.ebi.ac.uk/Tools/seqstats/emboss_pepstats/ accessed on 31 May 2020). The prediction of the transmembrane helices for the StLAX, StPIN, and StABCB proteins was performed by TMHHM v.2.0 (http://www.cbs.dtu.dk/services/TMHMM accessed on 31 May 2020). Additionally, protein subcellular localization was predicted by WoLF PSORT (http://www.genscript.com/psort/wolf_psort.html accessed on 31 May 2020).

### 2.2. Genome Distribution, Phylogenetic Tree Construction, and Promoter Analysis

The chromosomal location data of StLAX, StPIN, and StABCB family genes were obtained from phytozome 12.1.6. Distinctive gene names were arranged according to the position from the top to the bottom on chromosomes 1–12. In addition, visualization of chromosomes and segmental duplications were employed using TBtools to draw out the corresponding circos. Gene pairs with nucleotide sequence identities over 90% were considered as duplicated genes, which were analyzed by DNAMAN software. The alignment containing full-length amino acid sequences of LAX, PIN, and ABCB from Arabidopsis, rice, tomato, and potato was generated by ClustalW program with the default parameters, and the resulting sequence alignments were then uploaded to construct the unrooted neighbor-joining tree by the methods of *p*-distance and complete deletion, with a bootstrap of 1000 replicates using MEGA 6.0 (http://www.megasoftware.net/ accessed on 31 May 2020). Additionally, the promoters (2000 bp) of StLAX, StPIN, and StABCB genes were obtained from Phytozome 12.1.6. Auxin responsive and stress-related *cis*-regulatory elements analyses were performed using New PLACE (https://www.dna.affrc.go.jp/PLACE/?action=newplace accessed on 8 June 2020).

### 2.3. Weighted Co-Expression Network Construction

RNA-Seq data from different organs of *S. tuberosum* were chosen to construct a scale-free gene co-expression network. Gene expression levels were normalized to fragments per kilobase per million (FPKM) values. Network analysis was performed using the weighted gene co-expression network analysis (WGCNA) R software package following step-by-step network construction and the module detection method [50]. A proper power-law coefficient β was selected using the soft-thresholding method, and module stability was tested as the average correlation between the original connectivity and the connectivity from half samples that were randomly sampled 1000 times. The dynamic hierarchical tree cut algorithm was used to identify the co-expression gene modules based on the topological overlap matrix (TOM). Each module was summarized by a module eigengene (ME) through singular value decomposition. Additionally, a permutation test *p*-value was performed to estimate the correlation of differential expression of their corresponding eigengenes in the various tissues from potato plant, and a gene significance (GS) measure could also be defined by minus log of a *p*-value. Finally, a clustering diagram was plotted with the hclust function in the WGCNA package, and the heatmap for the correlation coefficient between modules was generated using the R package heatmap.3.R.

### 2.4. Gene Structure and Tissue-Specific Expression Profiling Analysis

The Gene Structure Display Server (GSDS) (http://gsds.cbi.pku.edu.cn/ accessed on 31 May 2020) was employed to identify exon–intron organizations of StLAX, StPIN, and StABCB family genes by comparing the coding sequences with their corresponding genomic sequences, which were collected from phytozome 12.1.6. To characterize the expression patterns of the StLAX, StPIN, and StABCB genes, we used the RNA-Seq data (DM_v4.03) of various tissues of the RH89-039-16 genotype (referred to as RH), including flower, leaf, shoot apex, stolon, young tuber, mature tuber, and root tissues downloaded from PGSC. The expression levels were calculated as FPKM values. The raw data of FPKM values were subsequently converted as log2 and then submitted to HemI [51] for establishing the expression heat maps of hierarchical clustering of the StLAX, StPIN, and StABCB genes.

### 2.5. Three-Dimensional (3D) Structure Predictions and Subcellular Localization of Auxin Transporters in S. tuberosum

The Phyre2 web portal was used for protein modeling, prediction, and analysis [52]. Three typical auxin transporters StLAX2, StPIN2, and StABCB4 were used for this analysis. The StLAX2-GFP vector, the StPIN2-GFP vector, and the StABCB4-GFP vector were constructed and transformed into Agrobacterium strain GV3101 by the freeze–thaw method. Transient expression in epidermal cells of *Nicotiana benthamiana* leaves was performed as described by Nie et al. (2019) [53], and the green fluorescence and the plasmolysis were detected by laser scanning confocal microscopy (LSCM). The GFP excitation wavelength was 488 nm, and the chloroplast autofluorescence excitation wavelength was 633nm.

### 2.6. Plant Growth, Treatments, and Collection of Tissues

The potato cultivar *Desiree* was used in this study. The plantlets were grown in Murashige and Skoog (MS) medium containing 2% sucrose and 0.8% agar and 0.05% MES (2-Morpholinoethanesulfonic Acid) at pH 5.8. All plantlets were grown in a plant growth chamber with a 16 h light (10,000 Lx) and 8 h dark (0 Lx) photoperiod at 22 ± 1 °C. Then the four-week-old plantlets were transferred into containers of 2% MS nutritional liquid medium again and sustained for four weeks under the same growth conditions as before. Potato plantlets with consistent growth vigor were subjected to the phytohormone and abiotic stress treatments. For hormone treatments, four-week-old plantlets were soaked in 2% MS nutritional liquid medium with 10 μM IAA, 50 μM 2,3,5-triiodobenzoic acid (TIBA), 30 μM 1-NOA, and 100 μM abscisic acid (ABA), then incubated for 3 h. For stress experiments, the roots of potato plantlets were immersed in nutritional liquid medium containing 200 mM NaCl or 20% (*w*/*w*) polyethylene glycol (PEG6000) for 24 h. Untreated plantlets were used as controls. The whole treated and control potato plantlets were collected for RNA extraction. For each treatment condition, three biological replicates were established to reduce the error rate, and a collection of samples from four potato plantlets were used as one biological replicate.

### 2.7. RNA Isolation and qRT-PCR Analysis

Total RNA from whole in vitro-grown plantlets was extracted using a high purity total RNA rapid extraction kit (TIANGEN, Beijing, China) based on manufacturer’s instructions. Gel electrophoresis was used to assess RNA quality and quantity. First-strand cDNA was synthesized from 2 ug of total RNA using the Fast Super RT Kit cDNA with gDNase (TIANGEN, Beijing, China). The primers sequences of individual gene families for qRT-PCR analysis were designed with Primer Premier 5 software [54] and their specificity of unique and appropriate cDNA segments was confirmed by uploading to the BLAST program (Table A1). qRT-PCR was performed on the Q7 Real-Time PCR System using 2xRealStar Green Fast Mixture (GenStar, Beijing, China) with elongation factor 1-α (ef1-α) as the internal reference gene for normalization of gene expression [55]. The reaction was carried out in a total volume of 10 μL containing 0.4 μL cDNA as template, 5 μL RealStar Green Fast Mixture (2×), 0.4 μL of each forward and reverse primer (10 μM), and RNase-free water up to 10 μL. In qRT-PCR experiments, the following thermal cycling conditions were applied: initial activation 95 °C for 2 min, then 40 cycles of 95 °C for 15 s, 55 °C for 15 s, and 72 °C for 19 s. A melting curve generated from 65 °C to 95 °C, with increments of 0.5 °C every 5 s, was performed to check specific amplification. The relative RNA levels of each gene were calculated from cycle threshold (C_T_) values according to the 2^−ΔΔCT^ method [56]. The data were analyzed using SPSS software (SPSS version 19.0, SPSS, Chicago, IL, USA), using descriptive statistical tests; one-way analysis of variance was used to evaluate the differences between treatments and control. Statistical significance was established at 0.05 and 0.01, respectively.

## 3. Results and Discussion

### 3.1. Genome-Wide Identification of Stlax, Stpin, and StABCB Auxin Transporter Genes in Potato

From the reference genome retrieved from phytozome 12.1.6 for *Arabidopsis thaliana*, the 4 AtAUX/LAXs, 8 AtPINs, and 22 AtABCB protein sequences were used as queries to perform the BLAST searches against the available potato protein sequence data (DM_v3.4_pep_nonredundant) of potato downloaded from PGSC. A total of 5 putative StLAXs and 22 StABCBs were identified from the potato genome, and they were named in accordance with their location order on the chromosomes, with the exception that the nomenclature of 10 StPIN genes were set according to sequence similarity to the *A. thaliana* PIN genes based on previous research [49]. Detailed information regarding 37 *S. tuberosum* putative auxin transporter-encoding genes, including gene names, locus IDs, open reading frame (ORF) lengths, exon numbers, chromosome locations, deduced polypeptide basic parameters, transmembrane helices, and subcellular localization predictions, are provided in Table 1.

The deduced StLAX proteins ranged from 468 (StLAX5) to 494 (StLAX1) amino acids in length, and possessed a MW ranging from 53.20 kDa (StLAX5) to 55.71 kDa (StLAX1) and pI between 7.9 (StLAX4) and 8.8 (StLAX5) (Table 1). The prediction for StLAX proteins in regard to their subcellular localization provided a foundation for further functional research. StLAX1 was predicted to be localized within the cytoplasm, while StLAX2–5 were predicted to be PM-localized. Furthermore, topology analysis conducted using TMHHM v.2.0 revealed that all of the StLAX proteins possessed 10 transmembrane helices, indicating that their core regions were highly conserved (Table 1, Figure A1A). Notably, membrane protein StLAX1 was predicted to be cytoplasm-localized, which could be supported by the fact that a nearest paralogue AtLAX2 was unable to be correctly targeted to the PM when ectopically expressed in epidermal cells [10]. This PM targeting defect of AtLAX2 and StLAX1 might be required of them for additional trafficking factors that are performed by subfunctionalization in other tissues.

The size of the ORF for StPIN proteins varied from 783bp (StPIN8) to 1965bp (StPIN4). The lengths of the corresponding proteins ranged from 260 to 654 amino acids, and they possessed 28.41 kDa to 71.29 kDa molecular masses and predicted pI values of 6.82 to 9.69. The number of transmembrane domains for StPIN proteins ranged from 5 to 9 (Table 1, Figure A1B). Five StPIN proteins (StPIN2, StPIN3, StPIN5, StPIN6, and StPIN8) were putatively localized within the PM, while StPIN1, StPIN7, and StPIN9 were localized within chloroplasts. Specifically, StPIN4 was predicted to be localized both in the chloroplasts and the PM, and StPIN10 was predicted to be vacuolar-localized. Various subcellular localization patterns might be indicative of the specific functions of the PIN gene family.

The protein properties of the StABCB family varied dramatically. These proteins ranged from 638 to 1527 amino acids in length, possessed 68.41 kDa to 167.89 kDa molecular masses, and exhibited pI values of 7.35 to 9.58. Additionally, their ORF lengths ranged from 1917bp to 4584bp. For most StABCBs, the conversed domain topology consisted of 2–12 transmembrane helices distributed at the N- and C-termini. The majority of StABCBs possessed a common domain composition, with the exception of three members (StABCB6, StABCB15, and StABCB19) that were missing one or both domains (Table 1, Figure A1C). With the exception of StABCB15 that was localized within the chloroplast, all the other StABCBs exhibited a possible localization to the PM. This implies that these ABCB proteins may act as basal auxin transporters; however, no specialized research has been performed to elaborate on their concrete functions during plant growth and development.

### 3.2. Chromosomal Distribution of S. tuberosum LAX, PIN, and ABCB Auxin Transporter Gene Families

To visualize the organization of each auxin transporter gene within the *S. tuberosum* genome, chromosomal mapping of 5 StLAXs, 10 StPINs, and 22 StABCBs were arranged based on the position of these genes on 12 different chromosomes (Table 1, Figure 1). The 37 genes were found to be unevenly distributed, where StLAX1–5 were located in an orderly arrangement on chromosomes 01, 09, 10, and 11, respectively. Chromosomes 01, 09, and 11 contained only one StLAX gene each, while StLAX3 and StLAX4 presented together on chromosome 10. For StPINs, 8 out of the 12 *S. tuberosum* chromosomes contained StPIN proteins, and no StPIN genes were located on the other four chromosomes (chromosome 08, 09, 11, and 12). Nearly all of chromosomes with the exception of chromosome 01, 04, and 10 contained StABCB genes, and these included four StABCB genes on chromosomes 03 and 12, three StABCBs on chromosomes 02 and 06, and two on chromosomes 07, 09, and 11. Chromosomes 05 and 08 contained only one StABCB gene each. Previous study on potato genome analysis presented evidence for at least two whole-genome duplication events [48]. However, analysis of percent ORF nucleotide and amino acid identities of StLAX, StPIN, and StABCB gene families showed no veritable duplicated gene pair shared over 90% identity in both levels (Table A2, Table A3 and Table A4). Additionally, the gene pairs in the duplicated regions were StLAX1–StLAX3, StLAX2–StLAX4, StPIN3–StPIN4, StPIN7–StPIN9 and StABCB1–StABCB4 (Figure 1). Tandem duplication was also important in the expansion of the StABCB gene family. Three tandems duplicated ABCB loci pairs (StABCB9–StABCB10, StABCB17–StABCB18, StABCB20–StABCB21) were present on chromosomes 06, 11, and 12, respectively. In contrast, neither the StLAX loci nor the StPIN loci were derived from tandem duplication.

### 3.3. Phylogenetic Analysis of LAX, PIN, and ABCB Proteins in Arabidopsis, Rice, Tomato, and Potato

To date, auxin transporter-encoding gene families in the model plant Arabidopsis have been widely studied for their biological functions in the context of development and responses to the environment [31,35,41,57,58,59]. Additionally, there is increasing evidence of the roles auxin transporters play in auxin-regulated development in monocotyledon and other dicotyledon species [45,60]. Therefore, investigation of the phylogenetic relationships of auxin transporter proteins from closely related species including Arabidopsis, rice, tomato, and potato will be helpful for understanding the putative biological functions of auxin transporter genes in potato. A total of 5 StLAX, 10 StPIN, and 22 StABCB genes, including 4 AtAUX/LAXs, 8 AtPINs, 22 AtABCB proteins (21 transcribed genes and 1 pseudogene) from Arabidopsis; 5 OsLAXs, 12 OsPINs, 22 OsABCBs from rice; and 5 SlLAXs, 10 SlPINs, 29 SlABCBs from tomato were uploaded to construct an unrooted neighbor-joining phylogenetic tree (Figure 2, Table A5). A phylogenetic tree of the LAX proteins revealed two distinct subfamilies, where potato StLAX proteins clustered more closely with the other dicots (tomato and Arabidopsis) compared with those OsLAXs from the monocotyledon species. StLAX2 and StLAX4 proteins were grouped with the closely related SlLAX1 and SlLAX4 proteins, as they possessed 100% amino acid identity (Figure 2A). Potato StLAX1 and StLAX3 proteins also shared high homology with Arabidopsis AtLAX2. Moreover, a paralogue gene pair was present between potato StLAX5 and Arabidopsis AtLAX3 proteins.

A previous study performed a detailed analysis of PIN protein structure, and their findings identified two predicted transmembrane domains linked by the central intracellular loop [61]. More precisely, those possessing long loops were defined as canonical PIN proteins, and truly noncanonical PIN proteins possessed shorter loops. These terms gradually replaced the “long” and “short” terms that were previously used to describe the structural features of these proteins. The phylogenetic topology structure of the PIN proteins revealed two major clades. The first, which contained the members with the typical canonical structure, was comprised of Arabidopsis AtPIN1 to AtPIN4, and AtPIN7 together with 6 potato PINs (StPIN1–4, StPIN7, and StPIN9), 6 tomato PINs (SlPIN1–4, SlPIN7, and SlPIN9), and four OsPIN1, OsPIN2, and two OsPIN10 proteins. There existed six PIN ortholog gene pairs between potato and tomato, StPIN1–4, StPIN7, StPIN9 and SlPIN1–4, SlPIN7, SlPIN9, respectively. Moreover, two paralog gene pairs (StPIN3 with StPIN4 and StPIN7 with StPIN9) existed in the potato PIN gene family. Additionally, four OsPIN1 copies and two OsPIN10 copies showed a closer evolutionary relationship, which coincided with the fact that the enlargement of the monocot PIN family relied on whole genome duplications and the retention of multiple copies of similar proteins [45] (Figure 2B). The other clade that possessed the noncanonical AtPIN5 was located on the same branch as StPIN5/SlPIN5, StPIN10/SlPIN10, and OsPIN5, and AtPIN8 was grouped with StPIN8 and SlPIN8 due to high sequence similarity. Additionally, PIN6 is unique among the Arabidopsis PIN proteins due to its dual localization at the PM and ER, and based on this, it could not be classified as a canonical PIN protein or a noncanonical PIN protein according to the results of the localization study. In our study, Arabidopsis AtPIN6 and potato StPIN6 proteins constituted a separate clade from the other noncanonical PIN proteins, and this was in accordance with results from a previous publication [32].

A third class of auxin transporters, ABCB, was reported to mediate almost all aspects of plant growth and development, and several of these proteins (ABCB1, ABCB4, ABCB14, ABCB15, ABCB19 and ABCB21) have been well characterized with regard to their distinct and overlapping functions in *A. thaliana* [62,63,64,65]. Phylogenetic analysis of the 95 ABCB proteins from *A. thaliana*, *Solanum lycopersicum*, *O. sativa*, and *S. tuberosum* genomes indicated that the presence of nine clusters, with potato member(s) in the seven groups (Figure 2C). In clade I, AtABCB7 and AtABCB9 were very similar at the sequence level but separated from other ortholog gene pairs (AtABCB3 with AtABCB5, AtABCB11 with AtABCB12, and AtABCB4 with AtABCB21) in Arabidopsis. In particular, there was no StABCB protein in potato that was clustered with any AtABCBs, while StABCBs exhibited high sequence similarity to SlABCB proteins from tomato. In contrast to clade I, AtABCB2/10, AtABCB13/14, and AtABCB6/20 were deeply branched in clade VI. Meanwhile, StABCB2 and StABCB16 shared significant homology to ABCB19 and ABCB1, respectively, in Arabidopsis, indicating possible roles for StABCB2/16 in auxin transport in plant developmental programs based on close protein sequence similarity to AtABCB19/1. Interestingly, as observed for Arabidopsis ABCBs in clade V, five AtABCB genes (AtABCB15–18 and AtABCB22) subclustered with themselves, and a number of the OsABCBs grouped in the same way, suggesting that duplication events occurred within these genes.

### 3.4. Weighted Co-Expression Network Analysis of S. tuberosum Genes

A gene expression correlation network was constructed with 25,216 differentially expressed genes in various potato tissues by the WGCNA method for describing the correlation patterns among genes across microarray samples [50]. A soft thresholding power of 10 with a scale-free model fitting index *R*^2^ > 0.688 was chosen to maximize scale-free topology and maintain a high mean connectivity (Figure 3A,B). Then, a dynamic hierarchical tree cut algorithm was constructed to identify stable gene clusters composed of the differentially expressed genes and labeled by unique colors below, including 18 co-expression modules that were named black (983 genes), blue (2705 genes), darkgreen (4172 genes), darkgrey (448 genes), darkmagenta (253 genes), darkred (2526 genes), darkturquoise (461 genes), green (2354 genes), grey60 (530 genes), magenta (2984 genes), paleturquoise (281 genes), pluml (1016 genes), saddlebrown (289 genes), sienna3 (432 genes), skyblue3 (185 genes), tan (1852 genes), turquoise (3445 genes), and violet (265 genes), and 35 probesets were added to the “grey” module for they were not grouped in any of the 18 modules (Figure 3C). Each module expression profile was summarized by a ME, which assembled the most representative gene expression in a module. To understand the consensus modules’ significance for biological traits, differential expression of the module corresponding eigengenes across the various potato plant tissues is shown as correlation and *p*-values (Figure 3D). Obviously, eigengenes could be characterized by their differential expression profiles in different plant tissues. For example, 26 out of 37 putative auxin transporter genes in potato investigated in our study were distributed in 10 out of 18 consensus modules (Table A6). Furthermore, the construction of the cluster dendrogram among module eigengenes resulted in several meta-modules, and the relationships between them were highly preserved (Figure 3E). For example, one meta-module (comprised of the black and darkgrey module eigengenes) was represented by StABCB9 and StABCB16 that were differentially expressed in leaves. Another meta-module (comprised of the darkgreen and grey60 module eigengenes) was represented by StPIN1/2/3 and StABCB3/22 that tended to be differentially expressed in shoot apex. Moreover, the blue and darkmagenta module eigengenes including StPIN5/6, StABCB1/5/6/8/15 and StLAX4 tended to be co-expressed in stolon. Thus, this analysis revealed that the meta-modules corresponded to a biologically meaningful characterization of modules and genes.

### 3.5. Gene Structure and Tissue-Specific Expression of StLAX, StPIN, and StABCB Family Genes

To investigate the similarity and diversity in the gene structures of StLAX, StPIN, and StABCB family genes, exon–intron structure analysis was performed by comparing the coding sequences with the genomic sequences. The exon number of StLAX genes was either seven or eight, indicating a conversed gene structure (Figure 4, Table 1). From the exon–intron organization of 10 StPIN genes, the structural patterns of seven canonical type PIN genes (StPIN1–4 and StPIN6–7) were highly conserved with 6–7 exons divided by 5–6 introns, while the number of exons in the other three noncanonical StPIN genes (StPIN5, StPIN8, and StPIN10) varied from 3 to 5 and exhibited a smaller gene size in contrast to these canonical StPINs. In the StABCB gene family, the gene annotation predicted 6 to 17 exons, implying a dramatic variation in the exon–intron structure. This variation in both StPINs and StABCBs was primarily due to divergent intron length, which was one of the predominant factors affecting the gene size.

Analysis of the gene expression profiles in different tissues/organs can be helpful for exploring their possible biological functions. Based on available RNA-Seq-generated expression data for potato genotype RH, a heat map was constructed to reveal the expression levels of StLAX, StPIN, and StABCB genes in eight tissues/organs that included flower, leaf, stem, root, stolon, young tuber, mature tuber, and shoot apex (Figure 4; Table A7). The results reveal that most transcripts from the StLAX, StPIN, and StABCB family genes were detectable in all selected tissues. Exceptions included a failure to detect StPIN10 and StABCB12 and low detection of StABCB8. In the StLAX gene family, StLAX2 was expressed in all tissues at a high level. The transcript level of StLAX1 was higher in the stolon and exhibited a tissue-specific expression pattern. The remaining StLAX genes were constitutively expressed in all tissues. Additionally, all StLAX genes exhibited the lowest expression in mature tubers. For StPINs, StPIN1 and StPIN4 were ubiquitously expressed, while five out of the remaining StPIN genes were preferentially expressed at a specific developmental stage. StPIN2 and StPIN3 were more highly expressed in young tuber than in other organs, and StPIN5 was abundantly expressed in the roots. Meanwhile, two homologous genes, StPIN7 and StPIN9, exhibited relatively high levels in the stolon tissue, indicating that functional redundancy existed among the PIN genes. Finally, expression analysis of the StABCB gene family revealed that relatively high expression of StABCB2 and StABCB16 was present in all tissues examined and that StABCB1, StABCB5, and StABCB6 were expressed in a root-preferential manner, implying that these genes might play specific roles in regulating the development of root organs. A number of genes (including StABCB4, StABCB7, StABCB10, StABCB11, StABCB13, StABCB17, StABCB18, and StABCB20–22) exhibited almost no expression or possessed relatively low levels in all the analyzed tissues. To summarize these three auxin transporter gene families, genes of the StLAX family were expressed at significantly higher levels than the other two gene families, suggesting that the StLAXs might be more important for these designated developmental stages in potato.

### 3.6. Analysis of StLAX, StPIN, and StABCB Protein Structures and Subcellular Location

Previous topology analysis indicated that StLAX proteins contained a highly conversed core region composed of 10 transmembrane helices, and the C-terminus was proline rich and the N-terminus was rich in acidic amino acid. The 3D structure prediction of StLAX2 protein revealed that 393 residues (81% of StLAX2 protein) were modeled with 100% confidence by the single highest scoring template (Figure 5A).

Published studies have identified and illustrated 10 members of PIN family in potato, and they possessed a typical structure composed by a central hydrophilic loop within two highly conversed hydrophobic domains located at N- and C-termini, respectively. Structural divergences between PIN proteins were defined by a conversed modular loop structure, and most PIN proteins had a full hydrophilic loop of four highly conversed regions HC1–HC4 [61]. Notably, StPIN5, StPIN8, and StPIN10 lacked those four regions. We selected StPIN2 for its 3D structure analysis and found that 150 residues (23% of StPIN2 protein) were modeled with 99.2% confidence by the single highest scoring template (Figure 5B).

Topology prediction of StABCB proteins exhibited a common domain architecture with four domains: a repeated nucleotide-binding domain (NBD) and transmembrane domain (TMD). Within these two conversed modules, a more variable linker domain connected the first NBD to the second TMD. The majority of StABCB proteins possessed a common domain composition;however, three members (StABCB6, StABCB15, and StABCB19) were missing one or more domains. The 3D structure analysis indicated that 1248 residues (99% of StABCB4) were modeled with 100% confidence by the single highest scoring template (Figure 5C).

Most auxin transporters reported in other species were verified as PM-localized channel proteins, performing a function in auxin transporting [66]. The transient expression of three typical auxin transporters in epidermal cells of *N. benthamiana* leaves indicated that the green fluorescence signals from the expressed StLAX2-GFP, StPIN2-GFP, and StABCB4-GFP fusion genes were all concentrated on the PM, confirming that their putative roles in regulating auxin flow toward the cell membrane (Figure 5D–F).

### 3.7. Auxin Regulation of the StLAX, StPIN, and StABCB Genes

Auxin plays critical roles in regulating plant growth and development via influencing auxin transporters and maintaining intracellular auxin homoeostasis and redistribution [67]. Exogenous IAA application might impact auxin stimulation and transport [30]. To determine how auxin transporters in potato responded to exogenous auxin treatment, we tested the expression levels of members of the StLAX, StPIN, and StABCB gene families in response to treatment with 10 μM IAA for 3 h using qRT-PCR. The results indicate that most genes were responsive to IAA treatment. In the StLAX family, StLAX1–3 exhibited a lower expression level compared to that of StLAX4, which exhibited a moderate increase. A total of 5 StPIN genes (StPIN1, StPIN2, StPIN5, StPIN7, and StPIN10) were upregulated, and the remaining StPINs were downregulated at different levels after IAA treatment. It should be noted that StPIN6 expression in particular strongly decreased. For the StABCB family, the majority of StABCBs exhibited a positive response to auxin, where StABCB1, StABCB3, and StABCB9 sharply upregulated and only four genes considerably reduced compared to controls (Figure 6; Table A8). Overall, several StLAX, StPIN, and StABCB genes were upregulated in response to exogenous IAA, suggesting roles for these genes in auxin-related biological progress. There has been little functional characterization of auxin transporters in potato to date. However, phylogenetic relationships with Arabidopsis could provide clues in regard to the functional identity of auxin transporter genes in potato. In Arabidopsis, the expression of LAX3 was itself auxin-inducible, and LAX3 therefore functioned to promote lateral root emergence, where LAX3-expressing cells became more efficient sinks for auxin [15]. The paralogue of AtLAX3 in potato, StLAX5, exhibited a downward trend after auxin treatment and may also be involved in root development. It has been reported that StPIN5 was predominantly expressed in the root [49]. Meanwhile, the functional role of the ortholog gene (AtPIN5) found in *A. thaliana* in auxin homoeostasis has been confirmed. Therefore, it was possible that StPIN5 functioned to maintain auxin homoeostasis in potato based on the close phylogenetic relationship between AtPIN5 and StPIN5, and current evidence suggested that StPIN5 was upregulated in shoots and downregulated in roots in response to auxin treatment [68]. In the ABCB family, AtABCB19 was identified as an IAA transporter with strong induction in response to exogenous auxin treatment [9], while the expression of its presumed ortholog in potato, StABCB2, was not upregulated by IAA, suggesting that there existed a variational relationship between these two proteins. This motivated us to explore the role of StABCBs in auxin transport. Our findings reveal the existence of a similar feedback mechanism that functioned in regulating the expression of auxin transporter genes in *S. tuberosum*.

### 3.8. Expression Analysis of StLAX, StPIN, and StABCB Genes in Response to PATIs Treatments

Directional PAT is a form of active intercellular transport mediated by influx and efflux carriers that control many important plant growth and developmental processes. It is known that PATIs—including auxin influx carrier inhibitor 1-NOA and two auxin efflux carrier inhibitors, NPA (1-Naphthylphthalamic acid) and TIBA—are major tools that can be used to explore auxin-dependent biological processes. Here, we conducted a PATI assay to investigate the transcriptional fluctuations of StLAX, StPIN, and StABCB under 1-NOA and TIBA treatments to allow us to gain insights into the effects of PATIs on auxin transporters (Figure 7a,b; Table A8). Our results reveal that most genes were induced by PATIs. Surprisingly, all StLAX genes were inhibited by 1-NOA due to downregulation. In contrast, most StPIN and StABCB family genes were insensitive to 1-NOA treatment, with the exception of StABCB1, which was upregulated dramatically in response to treatment with 1-NOA, and of StABCB7, StABCB8, StABCB21, and StABCB22, which exhibited a more moderate increase. For TIBA treatment, most StPIN and StABCB auxin efflux carrier family genes were downregulated or exhibited slight variations, with the exception of StPIN3 and StPIN9 and StABCB8 and StABCB22, which exhibited an increase in response to TIBA. However, the response of StLAX genes to TIBA treatment was irregular, where two of five StLAXs were upregulated and three were downregulated to varying degrees. Overall, the expression of auxin transporter genes might be blocked by the PATIs. Additionally, we excluded the use of an additionally polar auxin transport inhibitor, NPA, in our present study, as a previous publication had pointed out that NPA was invaluable for demonstrating the involvement of the auxin efflux carrier during PAT-mediated developmental processes [69], and TIBA was sensitive to a greater number of genes than was NPA in regard to auxin response genes in Arabidopsis [70].

### 3.9. Expression of StLAX, StPIN, and StABCB Genes in Response to ABA and Abiotic Stresses

ABA is known as the plant “stress hormone” and is of predominant significance due to the important role it plays in mediating both biotic and abiotic stress responses in plants [71]. Current evidence has confirmed the specific roles for auxin as a regulator of environmental adaptation in plants [72]. Additionally, the dynamic subcellular trafficking and polarity of PIN proteins were both regulated by a number of environmental responses, thus leading to a complex mechanism that integrated PINs and auxin distribution [8]. To address the possible involvement of *S. tuberosum* auxin transporters in response to abiotic stress at the transcriptional level, qRT-PCR was performed to investigate the expression fluctuations of 37 auxin transporter genes in response to ABA, salinity (NaCl), and drought (PEG) treatments compared to that in untreated plantlets grown in nutrient solution as controls. The data indicated that the majority of genes were responsive to the three stress treatments (Figure 8; Table A9). The majority of StLAX, StPIN, and StABCB genes, with the exception of StPIN2, StPIN5, StPIN8, and StABCB1, StABCB6, StABCB8, StABCB12, StABCB17, and StABCB18, were inhibited by ABA (Figure 8a). In response to salt treatment, 3 of 5 StLAXs (StLAX1, StLAX3, and StLAX5), 4 of 10StPINs (StPIN1, StPIN2, StPIN6, and StPIN9), and 5 of 22 StABCB genes (StABCB4, StABCB8, StABCB9, StABCB16, and StABCB21) were significantly downregulated compared to levels in the control. In contrast, the rest of the three family genes were all induced by the application of NaCl, and even one or more were strongly upregulated, including StLAX4, StPIN10, StABCB1, StABCB10, StABCB14, StABCB18, StABCB20, and StABCB22 (Figure 8b). In a similar manner, one-half of the *S. tuberosum* auxin transporter genes were repressed by PEG treatment, and most of them were notably downregulated (Figure 8c). Interestingly, the transcription of several StABCB family genes (StABCB1, StABCB11, StABCB14, StABCB15, StABCB18, and StABCB20) was dramatically upregulated in response to PEG treatment. Overall, almost all of the *S. tuberosum* auxin transporter genes were transcriptionally mediated by stress hormone ABA, salt, and/or drought treatments, implying that the function of these genes might be associated with abiotic stress responses and adaption. The differential expression profiles of StLAX, StPIN, and StABCB genes indicated that the abiotic stress adaptive mechanism of plants was highly complex. Despite this complexity, a clear relationship between auxin transporter genes and abiotic stresses has been reported in the model plant Arabidopsis. The auxin influx mutant *aux1* was sensitive to high salt, where stress-induced lateral root elongation was completely blocked [73]. Additionally, salt stress altered the expression and localization of the PIN2 protein, resulting in reduced gravity response of root growth in Arabidopsis [74]. To date, the effects of hormone and abiotic factors on ABCB gene expression have been reported partially and included the finding that ABA reduced the expression of AtABCB4 and its close homolog, AtABCB21 [62,75]. The expression of PGP19 was suppressed by the activation of phytochromes and cryptochromes in Arabidopsis [43]. In the present study, we highlighted the responsiveness of StLAX, StPIN, and StABCB auxin transporter genes to hormones and abiotic stresses. Based on this preliminary research, further functional characterization of each transporter against abiotic stress will be performed using overexpression/knock-out transgenic studies of auxin transporters to help to elucidate the regulatory mechanisms of auxin-abiotic stress signaling.

### 3.10. Analysis of cis-Regulatory Elements in StLAX, StPIN, and StABCB Gene Promoters

*Cis*-regulatory elements located within promoter regions play a decisive role in the transcriptional regulation of their target genes [76]. To identify potential regulatory relationships among auxin transporter genes in potato and *cis*-regulatory elements located in their promoter regions, we retrieved the putative promoter sequences (2000 bp upstream the 5′UTR region) of StLAXs, StPINs, and StABCBs from Phytozome 12.1.6 for use in scanning of designated *cis*-regulatory elements. According to the statistical results, a total of 610 auxin-responsive and stress-related *cis*-regulatory elements were detected in variable numbers (Figure 9; Table A10). Auxin-regulatory *cis*-elements—including AuxRE (TGTCTC), bZIP response elements (ZREs), Myb response elements (MREs), ABA responsive elements (ABREs), and other abiotic and biotic responsive elements—could account for the response of StLAXs, StPINs, and StABCBs to most external stimuli. Of these, almost all genes in the three families possessed one or more auxin-regulatory *cis*-elements in their promoter regions, and these elements may be associated with their role in auxin transport. Notably, biotic stress responsive elements (W box) occurred at a high frequency of 2 to 26 sites at each promoter, and this prompted us to speculate that all of 37 *S. tuberosum* auxin transporter genes may be involved in adaption to biotic stress. Additionally, we found that many auxin transporter genes in potato seemed to have similar *cis*-elements in their promoter regions, while their expression profiles differed from one another. Presumably, there was no correlation between occupancy patterns of *cis*-elements and gene expression profiles. Even in yeast, Gao et al. confirmed the same conclusion [77]. It could be that the *cis*-regulatory elements were targeted by several transcription factors, such as the ARF binding site (AuxRE:TGTCTC), the WRKYs binding site (W box:TTGAC/TGACT), the bZIPs binding site (ABRE:ACGTG), the bHLHs binding site (MYCR:CACATG), the MYBs binding site (MYBR:CTAACCA), and other homeodomain proteins [78], indicating that a complex regulatory mechanism controls a number of these transcription factors that co-mediate the expression of StLAX, StPIN, and StABCB genes.

## 4. Conclusions

In summary, we have provided comprehensive information on StLAX, StPIN, and StABCB auxin transporter gene families in potato, which included basic parameters, chromosomal distribution, phylogeny, co-expression network analysis, tissue-specific expression patterns, gene structure and subcellular localization, transcription analysis under exogenous hormone stimuli and abiotic stresses, and *cis*-regulatory element prediction. The responsiveness of StLAXs, StPINs, and StABCBs to auxin and PATIs implied their possible roles in mediating intercellular auxin homoeostasis and redistribution. Additionally, the differential expression levels of StLAX, StPIN, and StABCB genes in response to ABA and abiotic stresses (salt and drought) suggested that there were specific adaptive mechanisms on tolerance to various environmental stimuli. Promoter *cis*-regulatory element description analyses suggested that a number of *cis*-regulatory elements within the promoters of auxin transporter genes in potato were targeted by relevant transcription factors to respond to diverse stresses. We are confident that our results provide a foundation for a better understanding of auxin transport in potato, as we have demonstrated the biological significance of these families of genes in hormone signaling and adaption to environmental stresses.

## Figures and Tables

**Figure 1 biology-10-00127-f001:**
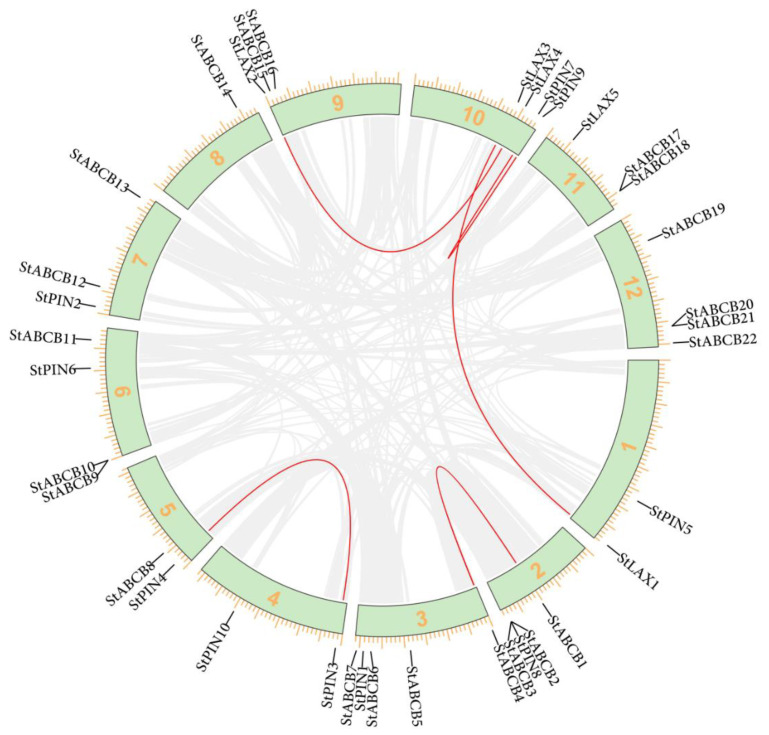
Chromosomal distribution of StLAX, StPIN, and StABCB family genes. Potato chromosomes are arranged in a circle. Five StLAX genes, 10 StPIN genes, and 22 StABCB genes are mapped by locus, and the duplicated gene clusters are represented by red lines.

**Figure 2 biology-10-00127-f002:**
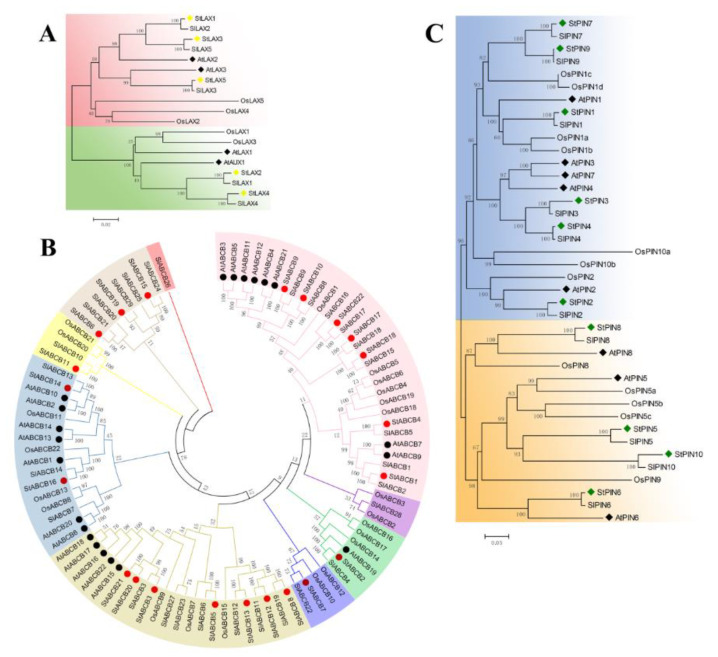
Phylogenetic tree of like auxin resistant 1s (LAXs) (**A**), pin-formed efflux carriers (PINs) (**B**), and ATP−binding cassette B influx/efflux carriers ABCBs (**C**) auxin transporter protein families in Arabidopsis, rice, tomato, and potato. Bootstrap values are presented for all branches. (**A**) LAX protein family: inventory of AtLAX families was based on TAIR databases. (**B**) PIN protein family: sequence data on AtPIN and StPIN families were based on TAIR annotation and Efstathios R’s publication. (**C**) ABCB protein family: inventory of AtABCB families was based on the ABC superfamily reviewed by Verrier et al. Different colors indicated different subfamilies.

**Figure 3 biology-10-00127-f003:**
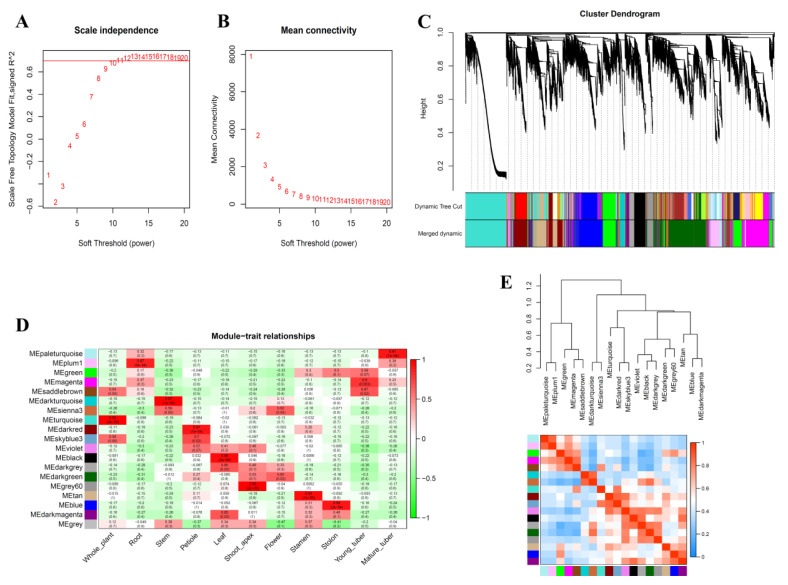
Graphical visualization of the *S. tuberosum* co−expression network. (**A**) Plot showing the scale free topology *R*^2^ value in function of increasing soft thresholding power. (**B**) Plot showing the relation between mean connectivity and soft threshold. (**C**) Dendrograms produced by average linkage of hierarchical clustering of *S. tuberosum* genes, which was based on a topological overlap matrix (TOM). The modules were assigned colors as indicated in the horizontal bar beneath the dendrogram. (**D**) Characterizing consensus modules by differential expression of their corresponding eigengenes in the various tissues from potato plant. Red means over−expression, green means underexpression; numbers in each cell give the corresponding *t*-test *p*-value. Each column corresponds to a tissue and each row corresponds to an eigengene. (**E**) Clustering dendrograms of consensus module eigengenes for identifying meta−modules (above) and the heatmap for the correlation coefficient between the modules (below). The diagonal plots show heatmap plots of eigengene adjacencies. Each row and column correspond to one eigengene (labeled by consensus module color). Within the heatmap, red indicates high adjacency (positive correlation) and green low adjacency (negative correlation), as shown by the color legend.

**Figure 4 biology-10-00127-f004:**
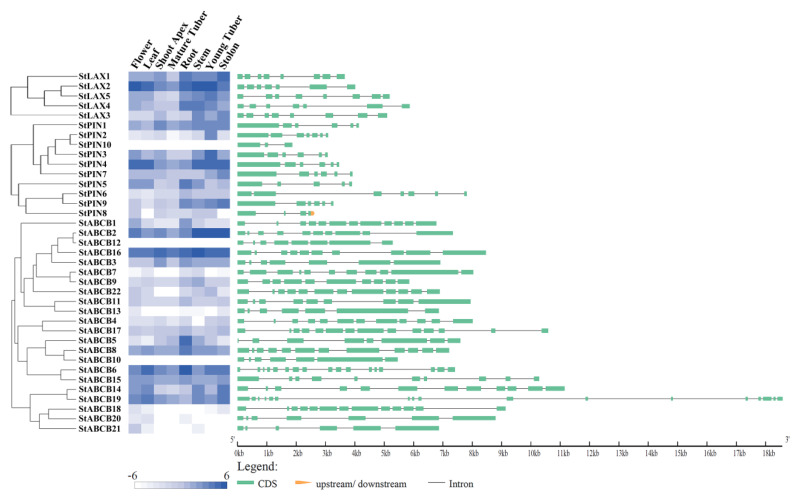
Tissues−specific expressions and exon−intron structures of StLAX, StPIN, and StABCB genes. The heat map was generated using the Cluster 3.0 software according to the RNA−Seq data of the RH89−039−16 genotype (RH). The exons are indicated by green boxes and the introns are indicated by gray lines.

**Figure 5 biology-10-00127-f005:**
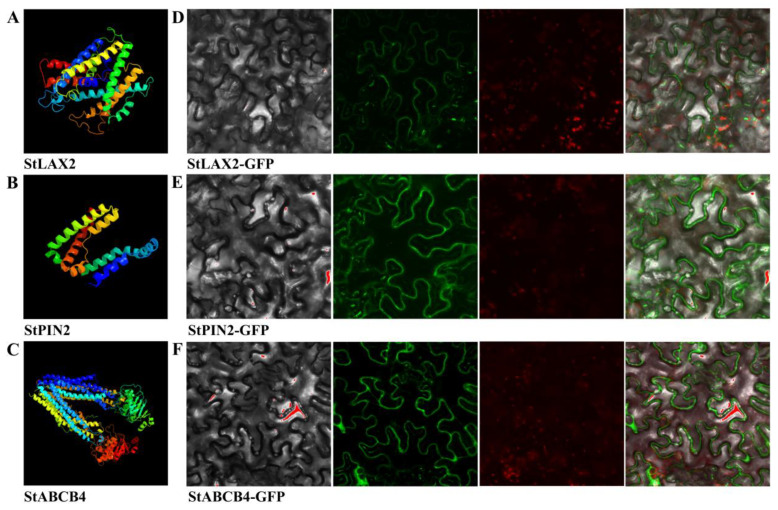
3D structure prediction and subcellular localization of *S. tuberosum* auxin transporter proteins. 3D structure of consensus sequences of three typical (**A**) StLAX2, (**B**) StPIN2, and (**C**) StABCB4 were used for this analysis. Auxin transporter protein−GFP fusion proteins (**D**) StLAX2−GFP, (**E**) StPIN2−GFP, and (**F**) StABCB4−GFP were transiently expressed in tobacco epidermis cells. Left to right: bright−field, green fluorescence of protein−GFP, chloroplast autofluorescence, and merged microscope images.

**Figure 6 biology-10-00127-f006:**
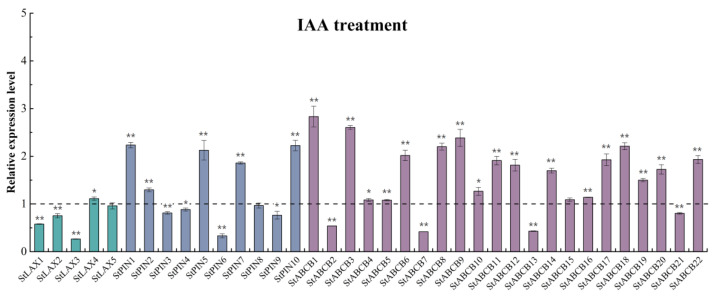
The relative expression values of auxin transporter StLAX, StPIN, and StABCB genes following indole−3−acetic acid (IAA) treatment. Total RNA was extracted from four−week−old potato plantlets for expression analysis. The histogram represented the relative RNA level of genes after IAA treatment compared with the mock expression level, which was normalized to a value of 1. The elongation factor 1−a (ef1−a) was employed as the internal standard to normalize the relative mRNA level of individual genes. Error bars represent the standard deviations (SDs) from three biological replicates. (* *t*-test *p*-value < 0.05, ** *t*-test *p*-value < 0.01).

**Figure 7 biology-10-00127-f007:**
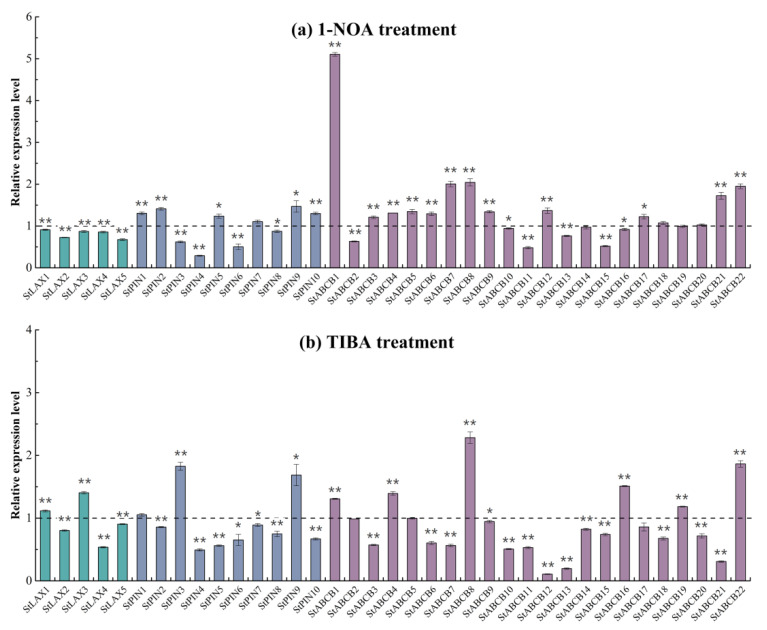
Expression profiles analysis of auxin transporter genes StLAX, StPIN, and StABCB under auxin transport inhibitor treatments. Potato plantlets grown at four−weeks−old were treated with 30 μM 1−naphthoxyaceticacids (1−NOA) (**a**) or 50 μM 2,3,5−triiodobenzoic acid (TIBA) (**b**) for 3 h. The relative expression levels were normalized to a value of 1 in the untreated seedlings. Assays were run in triplicate, and bars represent SDs. (* *t*-test *p*-value < 0.05, ** *t*-test *p*-value < 0.01.)

**Figure 8 biology-10-00127-f008:**
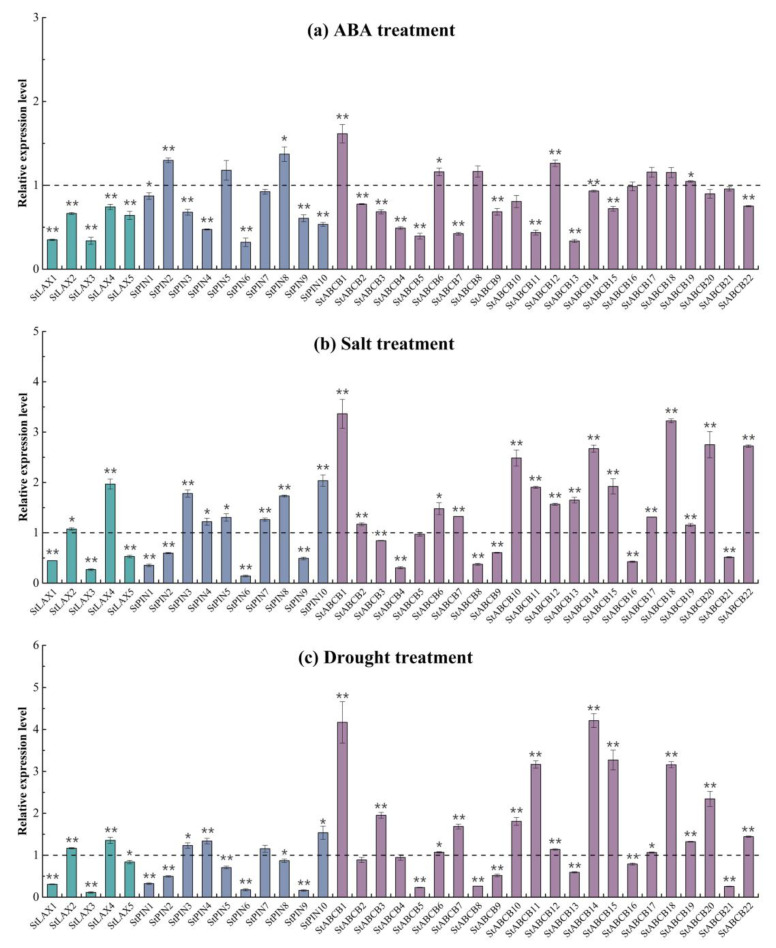
Expression levels of StLAX, StPIN, and StABCB family genes in response to abscisic acid (ABA) and abiotic stress. Total RNA was extracted from four−week−old potato plantlets treated with 100 μM ABA (**a**) for 3 h, 200 mM NaCl (**b**) or 20% (*w*/*v*) polyethylene glycol (PEG) (drought) (**c**) for 24 h. The relative mRNA level of each gene was normalized with respect to the internal reference gene (ef1−a). The data were analyzed by three biological repeats, and SDs are shown with error bars. (* *t*-test *p*-value < 0.05, ** *t*-test *p*-value < 0.01.)

**Figure 9 biology-10-00127-f009:**
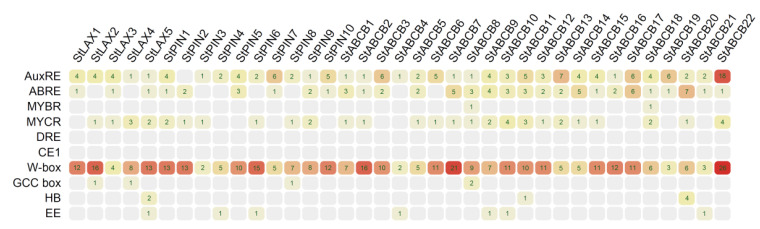
Analysis of auxin−responsive and stress−related *cis*−regulatory elements in the 2 kb promoter regions of StLAX, StPIN, and StABCB genes.

**Table 1 biology-10-00127-t001:** Information on StLAX, StPIN, and StABCB genes and properties of the deduced proteins in potato (*Solanum tuberosum*).

Gene ^a^	Locus ID ^a^	ORF Length (bp) ^a^	No.of Extrons	Chromosome Location (bp) ^a^	Deducted Polypeptid ^b^			No. of Transmembrane ^c^	Subcellular Localization ^d^
					Length (aa)	MI wt (Da)	pI		
*StLAX1*	PGSC0003DMT400004027	1485	8	ch01:87287332..87290998 (−)	494	55,706.95	8.5797	10	cyto
*StLAX2*	PGSC0003DMT400021923	1446	7	ch09:210800..214038 (−)	481	54,328.39	8.1707	10	PM
*StLAX3*	PGSC0003DMT400059693	1467	8	ch10:46747590..46751792 (−)	488	54,971.25	8.7078	10	PM
*StLAX4*	PGSC0003DMT400049377	1458	7	ch10:50480526..50485783 (−)	485	54,606.6	7.8706	10	PM
*StLAX5*	PGSC0003DMT400016760	1407	8	ch11:10042232..10046787 (−)	468	53,199.12	8.7862	10	PM
*StPIN1*	PGSC0003DMT400014752	1845	6	ch03:58350702..58354249 (+)	614	67,134.08	9.3381	8	chlo
*StPIN2*	PGSC0003DMT400048251	1896	7	ch07:2647114..2649884 (+)	631	68,666.48	9.4981	9	PM
*StPIN3*	PGSC0003DMT400015267	1395	6	ch04:2170473..2172957 (−)	601	65,908.59	7.0298	8	PM
*StPIN4*	PGSC0003DMT400078330	1965	6	ch05:4250058..4253070 (−)	654	71,290.07	7.3826	9	chlo/PM
*StPIN5*	PGSC0003DMT400046253	1068	5	ch01:64013966..64017139 (−)	355	39,264.82	9.2151	9	PM
*StPIN6*	PGSC0003DMT400079013	1587	7	ch06:41187368..41193747 (−)	528	57,652.42	8.8456	9	PM
*StPIN7*	PGSC0003DMT400072459	1764	6	ch10:57054506..57057784 (+)	587	63,906.07	8.8949	8	chlo
*StPIN8*	PGSC0003DMT400003569	783	4	ch02:46450539..46452728 (+)	260	28,407.25	9.6913	5	PM
*StPIN9*	PGSC0003DMT400021600	1785	6	ch10:59284672..59287550 (−)	594	64,304.33	9.4377	9	chlo
*StPIN10*	PGSC0003DMT400027309	966	3	ch04:49481160..49482767 (+)	321	35,909.45	7.2746	9	vacu
*StABCB1*	PGSC0003DMT400007960	3789	12	ch02:30568338..30574681 (+)	1262	136,961.15	7.3521	9	PM
*StABCB2*	PGSC0003DMT400003590	3750	10	ch02:46284463..46291628 (+)	1249	136,236.51	8.0288	9	PM
*StABCB3*	PGSC0003DMT400003546	3792	7	ch02:46615100..46621182 (+)	1263	137,488.42	8.0619	11	PM
*StABCB4*	PGSC0003DMT400034908	3780	12	ch03:822878..830598 (+)	1259	136,332.61	8.7596	10	PM
*StABCB5*	PGSC0003DMT400048379	3414	8	ch03:37579500..37585781 (+)	1137	124,433.16	8.2246	9	PM
*StABCB6*	PGSC0003DMT400063067	1917	17	ch03:55025184..55031533 (+)	638	68,411.61	8.8478	6	PM
*StABCB7*	PGSC0003DMT400058977	4584	11	ch03:61365905..61372730 (+)	1527	167,888.19	9.0608	12	PM
*StABCB8*	PGSC0003DMT400027962	3765	7	ch05:11042954..11047763 (−)	1254	137,231.31	9.1918	11	PM
*StABCB9*	PGSC0003DMT400018820	3864	12	ch06:336919..342710 (+)	1287	138,575.71	7.9275	9	PM
*StABCB10*	PGSC0003DMT400018812	3639	10	ch06:344414..349662 (+)	1212	130,723.67	7.799	8	PM
*StABCB11*	PGSC0003DMT400069516	3561	9	ch06:53569963..53576584 (+)	1186	130,633.57	7.4101	9	PM
*StABCB12*	PGSC0003DMT400013988	3681	8	ch07:11605714..11610849 (−)	1226	134,428.48	8.6851	11	PM
*StABCB13*	PGSC0003DMT400049576	3780	7	ch07:54289854..54295427 (+)	1259	137,937.76	9.2144	12	PM
*StABCB14*	PGSC0003DMT400045176	3774	12	ch08:49324870..49334350 (−)	1257	137,797.09	8.6611	10	PM
*StABCB15*	PGSC0003DMT400022893	1920	10	ch09:2609568..2618204 (+)	639	69,814.41	8.7886	2	chlo
*StABCB16*	PGSC0003DMT400009924	4002	10	ch09:5129170..5136543 (+)	1333	145,939.27	7.8141	11	PM
*StABCB17*	PGSC0003DMT400019156	3864	14	ch11:40225813..40235522 (+)	1287	141,384.58	9.5827	11	PM
*StABCB18*	PGSC0003DMT400019085	3885	13	ch11:40240421..40248719 (−)	1294	142,465.24	8.3389	12	PM
*StABCB19*	PGSC0003DMT400074962	2091	17	ch12:13618482..13637049 (−)	696	78,066.69	8.8706	4	PM
*StABCB20*	PGSC0003DMT400030345	3651	7	ch12:52311070..52319362 (+)	1216	133,255.89	8.8183	8	PM
*StABCB21*	PGSC0003DMT400030342	3096	6	ch12:52333684..52340071 (+)	1031	112,725.71	8.785	9	PM
*StABCB22*	PGSC0003DMT400011930	3864	12	ch12:59696990..59702667 (−)	1287	139,652.46	7.1317	11	PM

^a^ Gene information was retrieved from the *S. tuberosum* v4.03 genome annotation (phytozome 12.1.6: http://phytozome.jgi.doe.gov/pz/portal.html accessed on 31 May 2020). ^b^ Protein profiles were calculated using Pepstats (https://www.ebi.ac.uk/Tools/seqstats/emboss_pepstats/ accessed on 31 May 2020). ^c^ Transmembrane helices were predicted using TMHHM Server v2.0 (http://www.cbs.dtu.dk/services/TMHMM/ accessed on 31 May 2020). ^d^ Subcellular localization was predicted by WoLF PSORT (http://www.genscript.com/psort/wolf_psort.html accessed on 31 May 2020). ORF, open reading frame; PM, plasma membrane; cyto, cytoplasm; chlo, chloroplast; vacu, vacuolar.

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
