# Peer review of "Comprehensive Analysis and Expression Profiling of PIN, AUX/LAX, and ABCB Auxin Transporter Gene Families in Solanum tuberosum under Phytohormone Stimuli and Abiotic Stresses"

_biology, 2021, doi:10.3390/biology10020127_

Round 1
Reviewer 1 Report
The study of Yang and co-authors provides a comprehensive bioinformatics analysis of Solanum tuberosum auxin transporters. The article is complex - the bioinformatics analysis is enforced by experimental data, and well written. The authors have published previously similar research (Yang et al., 2019, Int.J.Mol.Sci), therefore having a good understanding of the topic and, importantly, of the methods (diversified and employed correctly) they use for this research.
I highly recommend the publication of their article. But the authors must firstly correct the references and their citation. I am convinced that the authors know very well the literature and probably there is a technical explanation for some references being messed up or incorrectly cited. Few examples of references cited incorrectly are at r.88 (to my knowledge pin1 phenotype has been described by Okada et al., 1991), r104 (pin5 by Mravec et al., 2009), r112 (pin8 by Ding et al., 2012), r383 (PIN phylogeny by Bennett et al., 2014), r701 (contains a conflicting reference). The reference list and the text citations must be carefully verified and corrected.
Other minor aspects that the authors may consider are:
-AUX/LAX paragraph from intro is written in the past tense instead present (e.g. AUX/LAX influx carriers encompassed ….)
-I would make clear already in the intro that PIN proteins are classified into two groups based on their loop length and localization at the ER or PM, and that they have distinct functions (polar intercellular transport and intracellular auxin homeostasis)
-I would abbreviate plasma membrane as PM (like in the text) and not plas (in the Table 1).
-Some images are so small that I could not read anything
-The authors mention in the text that most of PIN and ABCB genes are insensitive to NOA. This is not what the figure 7a shows – most of them actually responding
-I am not convinced about PIN3 missing the C-terminus. Most probably there is a wrong annotation of the gene sequence (such as ATG position) in the St data base.
Reviewer 2 Report
In this manuscript, the authors employed several approaches including data mining, phylogenetics, promoter analysis, gene expression through transcriptome/qRT-PCR, network analysis and hormone/stress experiments to identify a total of 37 auxin-transporter genes, their distribution in the potato genome, expression patterns, localization and the expression during hormone and stress treatment.
In general, I think this manuscript is sound. The results presented here will be useful for potato research community. I have some minor comments as below:
- Line 198: provide a reference to package WGCNA.
- Line 257: provide reference to Primer 5 software
- Line 275: how many sequences of each family were found in Arabidopsis and used for this analysis?
- For the WGCNA results, were there only 26 genes included in the identified DEGs (according to Table A6)? I think this should be clearly described in the results.
